# Origin, Function, and Implications of Intestinal and Hepatic Macrophages in the Pathogenesis of Alcohol-Associated Liver Disease

**DOI:** 10.3390/cells14030207

**Published:** 2025-01-30

**Authors:** Yifan Hu, Bernd Schnabl, Peter Stärkel

**Affiliations:** 1Laboratory of Hepato-Gastroenterology, Institute of Clinical and Experimental Research, Université Catholique de Louvain, 1200 Brussels, Belgium; yifan.hu@uclouvain.be; 2Department of Medicine, University of California San Diego, La Jolla, CA 92161, USA; beschnabl@health.ucsd.edu; 3Department of Medicine, VA San Diego Healthcare System, San Diego, CA 92161, USA; 4Cliniques Universitaires Saint-Luc, 1200 Brussels, Belgium

**Keywords:** alcohol use disorder, alcohol liver disease, intestine, liver, innate immune system, macrophage

## Abstract

Macrophages are members of the human innate immune system, and the majority reside in the liver. In recent years, they have been recognized as essential players in the maintenance of liver and intestinal homeostasis as well as key guardians of their respective immune systems, and they are increasingly being recognized as such. Paradoxically, they are also likely involved in chronic pathologies of the gastrointestinal tract and potentially in the alteration of the gut–liver axis in alcohol use disorder (AUD) and alcohol-associated liver disease (ALD). To date, the causal relationship between macrophages, the pathogenesis of ALD, and the immune dysregulation of the gut remains unclear. In this review, we will discuss our current understanding of the heterogeneity of intestinal and hepatic macrophages, their ontogeny, the potential factors that regulate their origin, and the evidence of how they are associated with the manifestation of chronic inflammation. We will also illustrate how the micro-environment of the intestine shapes the phenotypes and functionality of the macrophage compartment in both the intestines and liver and how they change during chronic alcohol abuse. Finally, we highlight the obstacles to current research and the prospects for this field.

## 1. Introduction

### 1.1. History of Alcohol Use

The first evidence of the consumption of alcohol as a drink dates back to 7000 BC in China, where human communities settled and agriculture started to flourish [1]. Evidence of the use of fermented rice, millet, grapes, and honey to make alcoholic beverages after this period has been found across the world, such as in Mesopotamia, ancient Egypt, and Syria [2]. Today, alcohol is largely consumed in all cultures as a social glue for recreational and ceremonial purposes because of its psychoactive, rewarding, and anxiolytic effects. However, the consumption of alcohol has begun to decline due to concerns over increased calories in peoples’ daily diets [3]. Culture and religious beliefs play the main role in the patterns of alcohol consumption in different regions and different races.

### 1.2. Epidemiology and Drinking Patterns

According to data from the World Health Organization (WHO), alcohol consumption varies significantly and depends on age and geographic location. The European region consumes more than 10.26 L of pure alcohol on average, which makes it the region that consumes alcohol the most. North Africa and the Middle East have the lowest alcohol consumption per capita (source: WHO data). According to the latest data, beer is the most consumed form of alcohol (396.76 billion L), followed by wine (25.76 billion L) and spirits (23.11 billion L) [4,5]. The total amount of alcohol consumed per capita has decreased by 5.5 L globally since 2015. In Europe, alcohol use has declined by 17% since 2000, whereas the use of alcohol in the Americas has been stagnating. By contrast, alcohol consumption has been steadily increasing in the Western Pacific and South-East by 40% and 112%, respectively. Alcohol use disorder (AUD) remains a significant issue worldwide. Although the exact circumstances that cause AUD to occur are not fully understood, several factors contribute to its development, such as environmental factors, genetic predisposition, peer interaction, and certain personality disorders.

### 1.3. Alcohol-Associated Liver Disease

Alcohol abuse is a major risk factor for multiple preventable causes of death and diseases such as alcohol addiction, alcohol-associated liver disease (ALD), diabetes, cancers, and fetal alcohol syndrome [6]. ALD contributes to 10% of deaths globally. ALD consists of a broad spectrum of disorders ranging from isolated steatosis to alcohol-associated steatohepatitis and cirrhosis. Although the underlying mechanisms remain poorly understood, the prevalence of patients with a history of chronic heavy alcohol intake that develop advanced fibrosis is around 30–35% [7]. In subjects with an AUD, the prevalence of alcohol-related hepatitis (AH), which is an acute deterioration that occurs in people with an underlying ALD, has been estimated to be 20–40% [8]. AH encompasses steatohepatitis, pericellular fibrosis, hepatocellular injury, and bilirubinostasis.

A previous study showed that the number of intestinal macrophages in AUD patients is significantly lower than in control subjects [9]; however, the reason behind this remains unclear. Overall, the study of macrophages is heavily based on animal models. Due to the fundamental differences in terms of anatomy, physiology, and metabolism between humans and other animals, many findings cannot be readily translated into clinical significance. As it is important to understand the role of macrophages in alcohol-associated liver disease, this review will stratify the current information, focus on the available clinical findings of how macrophages could participate in the pathogenesis of ALD, and highlight the obstacles and perspectives for future studies.

## 2. Macrophage Heterogeneity

Macrophages were first discovered by Metchnikoff in the 19th century and later classified as a part of the mononuclear phagocyte system along with monocytes and dendritic cells [10,11]. The early concept of mononuclear phagocytes led people to believe that the tissue-resident macrophages in adulthood rely on monocytes from peripheral blood to replenish the population. However, this belief about an inherent developmental link between tissue-resident macrophages and blood monocytes has been disproved. It is now well accepted that the long-living tissue-resident macrophages rely on in situ self-renewal. Moreover, certain macrophage populations in the embryonic stage are established before the emergence of circulating monocytes. The final discovery that reshaped the macrophage pedigree came from monocyte-specific knockout mice that demonstrated seemingly normal tissue macrophage compartments [12,13] despite the inability of the bone marrow to generate monocytes. Moreover, tissue macrophage numbers remain unaffected in human patients with monocytopenia [14]. These experiments and observations suggested an alternative, non-monocytic origin of tissue-resident macrophages and therefore led to a revision of the previous monocyte–macrophage axis. Current concepts postulate that yolk sac (YS)-derived macrophages are the main precursors for most tissue-resident macrophages. Fate-mapping studies have shown that late erythron-myeloid precursors that arise in the YS are the principal source of tissue-resident macrophages involving a monocytic intermediate. Alternatively, it has been demonstrated that fetal hematopoietic stem cells might also contribute to tissue-resident macrophage populations [14].

## 3. The Dynamics of Intestinal Macrophages

The specific cause of the variation in macrophage origins within different tissues is still not fully understood. Specifically, it is not yet clear why and how macrophages derived from embryos continue to exist in the central nervous system and epidermis, while their presence in substantial quantities does not persist in the intestinal mucosa in adulthood. Intestinal macrophages have a limited capacity for self-maintenance, and their expected half-life does not exceed 6 weeks. They are primarily replenished by bone-marrow- derived monocytes, which are CCR2-dependent [15]. One of many hypotheses in this regard is related to niche accessibility and availability. Indeed, the mucosa can be easily reached at all times, while the brain and epidermis are isolated from vasculature by the blood–brain barrier and the basal membrane, respectively. On the other hand, tissue macrophages that are not separated from the vasculature by a basal membrane, such as Kupffer cells and microglia, also function autonomously, which may indicate that proximity to the tissue might not be the reason behind both the monocyte infiltration of tissues and the reinforcement of the local immune system [16,17]. New evidence has indicated that microbiota and diet are capable of modifying macrophage profiles and how they both influence the dynamics of mucosal macrophages. Studies have illustrated that germ-free and SPF mice have a significantly lower number of intestinal macrophages [18,19,20]. The administration of antibiotics can reduce macrophage numbers, suggesting that the microbiota plays a role in augmenting macrophage turnover [21]. Gut dysbiosis, the intestinal mucosal microenvironment, dietary metabolites, and mechanical stress such as cyclic stretching have been suggested to play roles in upregulating pro-inflammatory cytokine release [22]. These gut-produced inflammatory signals potentially exacerbate monocyte recruitment and macrophage activation [23]. Additionally, the presence of commensal microbiota in the colon significantly alters macrophages’ gene expression profiles and surface markers promoting the development of CD11c+CD121b+ and CD11c−CD206^hi^ macrophages, particularly when the differentiation of monocytes starts. A differential turnover rate of tissue macrophages has been observed in different anatomical locations of the gut wall and in the liver, which could indicate that particular niches do not support macrophage self-renewal. However, differential rates of tissue macrophage subpopulations have not been assessed experimentally. Although progress has been made in understanding the dynamics of macrophage turnover, more tailored projects are still needed to fully understand this process and the functions of different subtypes of macrophages.

### 3.1. Function of Gut Macrophages

Like their counterparts in the liver, intestinal macrophages are involved in pathogen clearance. Consistent with their phagocytic nature, intestinal macrophages express many phagocytosis genes, such as Mertk, cd36, cd206, ltgav, ltgb5, and many others [24,25]. Macrophages also actively participate in tissue remodeling and apoptotic cell clearance, known as efferocytosis [25], which is mediated by integrin αvβ3. They also play a role in various activities implicated in maintaining gut homeostasis (Figure 1). Although intestinal macrophages are avidly phagocytic, they do not lead to overt inflammation in response to the commensal gut microbiome, which mainly occurs through the regulation of surface-receptor expression and reduced secretion of pro-inflammatory cytokines such as CCL2 and IL-8 [26,27]. Intestinal macrophages can also produce IL-10 (CX3CR1hi intestinal macrophages) and transforming growth factor β (TGF-β) [27], two prominent cytokines that predominantly contribute to reduced inflammation and homeostasis of the GI tract. IL-10 from various sources is sensed via the IL-10 receptor (IL-10R) to inhibit the synthesis of proinflammatory cytokines [27]. IL-10 not only limits the inflammatory state of macrophages but also regulates the killing of intracellular bacteria by those cells. Human-induced pluripotent-stem-cell-derived macrophages without a functional IL-10R display impaired bacterial killing due to dysregulated PGE2 production [28].

The sub-epithelial positioning of lamina propria (LP) macrophages indicates that they are placed to capture any foreign substance that penetrates the epithelial barrier. Murine models show that LP macrophages can also form transepithelial dendrites (TEDs) to sample luminal bacteria, constituting a cellular process that allows macrophages to penetrate the gut barrier without disrupting the tight junctions and the integrity of the epithelium. This process is highly dependent on the CX3CL1-CX3CR1 axis [29,30,31] (Figure 1). In particular, when encountering Gram-negative bacteria, intestinal macrophages downregulate expression of CD14, a glycoprotein receptor for lipopolysaccharide (LPS), compared to the levels expressed by conventional macrophages to reduce the inflammatory response [32]. Similarly, it has been noticed that CX3CR1 macrophages in the upper intestine are capable of capturing food particles, and it is hypothesized that they are involved in oral tolerance to food antigens [33].

However, in inflammatory conditions, the mature intestinal macrophages, along with other cells, release pro-inflammatory cytokines and monocyte chemotactic proteins such as CCL2, CCL7, CCL8, and CCL12 [34,35,36]. In chronic alcohol abuse conditions, the upregulation of CCL3 and CCL4 is especially common in the gut [37]. The consumption of alcohol increases gut permeability, disrupts intestinal tight junctions, and promotes microbial translocation [38]. In addition, chronic alcohol exposure directly alters the population and composition of the microbiota by decreasing the population of Gram-positive bacteria and promoting a number of Gram-negative bacteria [39,40].

Among the macrophages located in the lamina propria, it has been suggested that CD11c+ macrophages are monocyte-derived short-lived macrophages [27]. Research based on mice chronically given alcohol has also indicated that CD11c+ macrophages are concentrated close to the tips of the villi and express a pro-inflammatory profile, whereas CD206+ macrophages are located at the bases of crypts and fulfill the function of tissue macrophages. These mechanisms likely contribute to the dysregulation of macrophage numbers and macrophage immunoregulatory functions in AUD patients.

### 3.2. Gut Macrophages and Crosstalk with Other Immune Cells

The induction of gut-specific T-reg cells plays an important role in gut homeostasis [41]. These cells are regulatory cells for immune responses and have homing properties in the mesenteric lymph nodes [42,43] (Figure 1). It is believed that this process also involves macrophage-secreted IL-10 since Cx3cr1-induced deletion of IL-10 leads to a reduction in the frequency of antigen-specific Tregs. On the contrary, the deletion of IL-10 in macrophages does not have an observable impact on the global abundance of Tregs [44,45]. A murine model was used to demonstrate that the deletion of IL-10R hinders the differentiation of monocytes into pro-fibrogenic macrophages, which ultimately leads to an increase in the number of pro-inflammatory macrophages in the intestine. Additionally, macrophages can secrete IL-1β to facilitate the induction and maintenance of commensal-specific Th17 T cells [46,47]. On the other hand, T-cell-derived IL-10 is required for the regulation of intestinal homeostasis by the T cell–macrophage IL-10 axis (Figure 1). Indeed, appropriate IL-10 signaling levels are important for regulating different macrophage functions [28].

Compared to conventional dendritic cells (cDCs), macrophages are scarcely involved in the activation of naïve T cells [48]. However, high expression of MCH class II could mean that they are more likely to be involved in antigen presentation for previously activated T cells in the liver and intestines. Macrophages may influence T cell priming by modulating the cDC differentiation process. For example, macrophage IL-1β has been demonstrated to enhance CSF2 release by type 3 innate lymphoid cells (ILC3) [49], which is the principal cytokine for cDC differentiation in the tissue. The upregulation of CSF2 in macrophages is also a common hallmark for chronic-alcohol-abuse-induced inflammation in ALD [50]. However, most of the findings regarding ALD and tissue macrophages (see below) are largely based on murine or other animal models, and the exact mechanisms and immune responses still need to be clarified in humans.

### 3.3. Gut Macrophage Crosstalk with Enteric Nervous System

While it has been known for many years that macrophages are present in deeper layers of the gut wall [51], only recently has work begun to investigate their roles in intestinal homeostasis. Macrophages in the muscularis are intimately associated with the enteric nervous system and, in mice, appear to be morphologically and transcriptionally distinct [52]. Apart from LP macrophages, the macrophages in the muscularis also participate in bidirectional cross talk with enteric neurons. When an enteric neuron expressing the BMP receptor is activated by the macrophage-derived bone morphogenic protein 2 (BMP2), it starts producing CSF1, which can maintain the muscularis macrophage pool and further BMP2 expression [53,54] (Figure 1). These multi-system interactions regulate enteric smooth muscle contraction, thereby controlling peristalsis. This process can be disrupted by the administration of broad-spectrum antibiotics [20], which indirectly suggests that the microbiota plays a role in intestine contraction to a certain extent [52].

By generating metabolites of alcohol and bacteria-derived products, the microbiota also contributes to immunological changes in the intestine affecting the various macrophage niches. A study using a murine model has shown that one of the consequences of alcohol use is that the gut microbiota produces neuroactive metabolites that can trigger vagus nerve activation and directly excite the central nervous system via the systemic circulation or, alternatively, through the modulation of the immune system, e.g., via altering the macrophage population associated with the enteric nervous system [53]. Furthermore, intestinal macrophages might also serve as signal-transmitting cells in neuro-immune-stem-cell regulatory circuits, modulating intestinal stem cell self-renewal via upstream enteric serotonergic neurons. This process implicates the production of the neurotransmitter serotonin in enteric neurons, a process that can be promoted by gut microbiota metabolite valeric acid.

Overall, intestinal macrophages, which initially originate from embryonic stem cells, play a role in maintaining gut homeostasis and carry out important immunoregulatory functions towards commensal microbes while also protecting against invading pathogens. Given their limited half-life, their renewal depends on bone-marrow-derived monocytes. Intestinal macrophages can modulate the overall immune state by sampling the luminal microbiome to prevent excess inflammation. It is likely that macrophages located in the various niches of the intestine (villi, crypts, muscularis mucosae, the vascular bed, etc.) have different functions.

As the sentinels of the GI tract, intestinal macrophages extensively communicate not only with other immune cells but also neurons and the epithelium to ensure gut barrier integrity. Chronic alcohol consumption interferes with the gut microbiota, the epithelium, and the intestinal immune system. However, how alcohol disrupts physiological processes in the gut in AUD patients is not fully understood.

## 4. Origin and Composition of Hepatic Macrophages

The liver encounters constant blood flow from arterial circulation and the portal vein. Hence, the liver plays a vital role in host defense. It hosts the majority of all the macrophages in the body and is patrolled by circulating monocytes [54]. Kupffer cells (KCs) are liver-resident macrophages that originate from yolk sac progenitors during embryonic development in mammals. Although Kupffer cells reside in the liver, they are not primarily found in liver parenchyma. Indeed, the blood stream goes through the liver parenchyma using a network of tiny vessels, the sinusoids, where the KCs are located. This anatomical signature means that the liver is constantly exposed to microbial products, xenobiotics, and antigens from the GI tract and ensures effective antigen capture and presentation in the liver microenvironment [55]. In the absence of a liver injury, the macrophage pool primarily contains KCs, which have the ability to engage in self-renewal from intrahepatic precursors and, to a much lesser extent, bone marrow (BM)-derived macrophages [56,57,58]. However, the situation changes when a liver injury occurs. Some data suggest that severe liver injuries are associated with depletion of KCs. This depletion creates an intrahepatic niche that allows monocytes to infiltrate the liver. After 2–8 weeks, the infiltrated monocytes and monocyte-derived macrophages slowly adopt the transcriptional profile of KCs and become long-lived tissue-resident macrophages [56,59]. In the steady stage, KCs comprise the vast majority of the hepatic macrophages, yet another population of macrophages with a dendritic cell morphology that reside in the liver capsule has been identified in an animal model [60]. These cells are phenotypically and developmentally different from KCs but function as macrophages in the event of a liver injury to facilitate inflammatory responses (Figure 2).

### Hepatic Macrophages and Crosstalk with the Gut Microbiota

Much like gut macrophages, KCs are directly influenced by the gut since food metabolites and bacterial products are passed from the gut to the liver through the portal vein. They can modulate immune tolerance under physiological conditions [28] and are able to eliminate a small number of bacterial products translocated from the gut to the liver under physiological conditions without triggering inflammation in the liver [59,61]. This delicate homeostasis can be disrupted at different levels, such as via alterations in the gut microbiome, increased gut permeability, excessive exposure of the liver to microbial products, and alteration of hepatic macrophage activity, leading to a pro-inflammatory environment in the liver. Among all the molecules that can influence the polarization of hepatic macrophages, Gram-negative-bacteria-derived LPS, which is delivered to the liver via portal blood circulation, is constantly stimulating macrophages. Hepatic macrophages are able to clear LPS in circulating blood without being activated themselves, constituting a phenomenon called LPS tolerance. Some studies have shown that LPS-TLR4/MD2-TNFα signaling on hepatic macrophages via sensing endotoxin translocation induces various inflammation responses [62,63,64]. In addition, receptor-interacting protein kinase 3 (RIP3), regulated by the gut microbiota and endotoxin translocation, also plays an important role in the pathogenesis of liver inflammation. It has been suggested that in alcohol-associated liver disease, RIP3 is important to hepatic macrophage pro-inflammatory cytokines’ responses and that the absence of RIP3 inhibits alcohol-induced liver injury [65,66,67]. Indeed, it has been shown that CGIg+ (complement receptor of the immunoglobulin superfamily) macrophages are responsible for the clearance of microbial DNA and pathobionts and that alcohol consumption reduces microbial DNA clearance by Kupffer cells, which in turn exacerbates liver inflammation and damage [68,69]. In AUD and ALD patients, there is a rise in systemic endotoxin levels together with an increase in sensitivity to LPS [70,71]. The traditional proposition in this regard is that in the progression of ALD, the constant exposure of LPS converts KCs from an immune tolerance state to an immune activation state [61,72,73]. However, recent findings instead suggest that hepatic macrophages behave as a whole pool, with KCs maintaining their immune tolerance, but monocyte-derived macrophages are either proinflammatory or profibrogenic [74,75]. Moreover, it has been postulated that the activation of monocytes/macrophages not only depends on signals from the gut microbiota but also on endogenous messengers from the hepatic microenvironment [61,76]. The dogma behind the different roles of KCs and monocytes/macrophages still needs to be further investigated.

Hepatic-tissue-resident macrophages, Kupffer cells, primarily derive from the YS. In the steady state, the liver can be considered a closed system with slow macrophage turnover and a negligible need for replacement in adulthood. Kupffer cells are constantly exposed to low levels of food particles and microbiome derivatives delivered by the portal vein from the GI tract without necessarily inducing liver inflammation. It is not surprising that hepatic macrophages are reshaped by the gut microbiome. Under pathological conditions, monocytes might infiltrate the liver and transform into pro-inflammatory macrophages. In the context of chronic alcohol abuse, where the composition of the gut microbiome is changed and gut permeability is increased, Kuppfer cells might adopt a different, pro-inflammatory phenotype in ALD patients, thus contributing to disease progression. 

## 5. Identifying Macrophages in the Liver and Intestines

One of the major issues that has stifled our progress in understanding the immunology and functionality of hepatic and intestinal macrophages is their inaccurate identification. For example, although in murine models macrophages have been identified based on the expression of the pan-macrophage marker F4/80 [77], it has been proven that cDC and eosinophiles can express F4/80 to some extent [77,78]. In human studies, CD68 is used as a pan-macrophage marker, but a cytological study has shown that non-myeloid cells, such as fibroblasts, can express less-abundant CD68 to some extent [79]. Moreover, many macrophages, including those in the liver and especially in the intestine, express high levels of CD11c and MCH-II, which have been used as classical markers for identifying dendritic cells [80,81,82]. Thus, the identification and subsequent phenotyping of hepatic and intestinal macrophages require a multi-parameter approach. CD64 (FcγR1) and Mer tyrosine kinase (MerTK) have emerged as the most promising markers for identifying tissue macrophages [83,84,85]. CD64 can also be used across species as a macrophage marker. More recently, in an immunology study, CD163L was proposed as the marker for tissue-resident macrophages both in lymphoid tissue and in the colon, and it was determined that CLEC5A may be used as a monocyte-derived pro-inflammatory macrophage marker. However, the results are not entirely conclusive [86]. When applying the combination of CD11c and MCH-II, the expression of CD64 can distinguish bona fide cDC in the intestine [84,87,88]. CD64 cannot be the only macrophage marker used since CD64+ macrophages are dependent on colony stimulating factor 1 (CSF1), which is released by enteric neurons, whereas CD64− CD11c+ MHC-II+ Mϕs are highly dependent on the cDC-specific growth factor Flt3L [87,88]. Some research has shown that CD64− CD11c+MHC-II+ Mϕs constantly migrate to mesenteric lymph nodes to maintain and activate T cells, a CCR7-dependent process [87,89,90,91]. However, in the intestine, CD64+ Mϕs cannot migrate into adjacent tissues, [27,48,92]. In a Cx3cr1+/gfp knock-in murine model, CX3CR1hi macrophages can be observed and localized in the gut, including in the submucosa, where mature macrophages express high levels of CD11b, and the submucosal muscularis [52,53,93,94,95,96].

A set of studies on multiple tissue macrophages using multiple analysis approaches has demonstrated an alignment of tissue macrophages between humans and mice. The use of markers such as CD14 and CD64 suggests that monocytic macrophage markers are preserved across species [26,36,78,91]. The expression of CD4, CD163, CD172a, and CD206 is conserved between humans and mice [24,26,27,97,98]. Despite the presence of major similarities, there are important differences between murine and human intestinal macrophages. The major difference is that human mature intestinal macrophages only express low levels of CX3CR1 and CD11c, unlike their murine counterparts. Another work has demonstrated phenotypic heterogeneity between macrophages in the lamina propria and muscularis, where muscularis macrophages express higher levels of CD14 and CD11b compared to the ones in the lamina propria [32].

Blood monocytes, the circulating precursors of macrophages and dendritic cells, have subtypes as well, which are characterized by different CD14 expression levels in humans. CD14^hi^ CD16− classical monocytes express the chemokine receptor CCR2 and rapidly infiltrate tissue upon injury, while CD14^low^ CD16^hi^ monocytes express higher levels of CX3CR1 and exhibit a patrolling behavior in vivo [99]. Classical and patrolling monocytes crawl along the liver endothelium, and it is possible that they exhibit this behavior to orchestrate the disposal of apoptotic cells and debris. However, this behavior is not observed in intermediate monocytes, indicating that monocyte migration is type-specific and dependent on adhesive molecules [100].

## 6. Monocytes and Macrophages in Alcohol Use Disorder and Alcohol-Induced Liver Disease

### 6.1. Implication in Intestinal and Hepatic Inflammation

Circulating monocytes are the precursors of macrophages and dendritic cells. Compared to their homeostatic counterparts, the classical CD14^hi^ monocytes in the liver produce a wider range of pro-inflammatory cytokines and chemokines, such as TNFα, IL1β, IL-6, IL-8 MCP-1, and ROS [101,102,103,104]. By contrast, in the gut, these macrophages produce a wider range of pro-inflammatory cytokines and chemokines, like TNFα, IL-6, IL1β, IL-8, TLR-2, TLR-4, CX3CR1, and MCP-1-4 [104,105,106], upon stimulation of toll-like receptors (TLR) by their appropriate ligands, mirroring the observation in IDB studies on humans [27,84,93]. Patients with alcohol use disorder show increased monocyte numbers, especially with respect to intermediate and non-classical monocytes [104,107] (Figure 3). Serum levels of LPS are positively correlated with circulating pro-inflammatory cytokines, which further indicates the activation of monocytes occurs in a TLR-dependent manner [71]. Apart from the existing evidence of the role of TLR-4 in ALD, it has been demonstrated that TLR-2 plays an important role in monocyte activation during the inflammatory response induced by alcohol. The activation of TLR-2, which is dependent on peptidoglycan but not LPS, is associated with an exacerbation of liver damage [104]. In animal models of chronic alcohol exposure, an influx of monocytes often indicates tissue injury and subsequent disease [108]. However, in the setting of binge drinking, monocyte TLR-4 expression is rapidly downregulated, whereas systemic IL-1β levels are increased. A similar upregulation was observed after the ex vivo LPS stimulation of CD14+ monocytes obtained from control, non-alcohol-fed mice. The levels of circulating non-classical monocytes, which are considered anti-inflammatory, are also elevated. After 24 h of alcohol abstinence, the serum level of IL-1β and the number of non-classical monocytes return to their baseline levels [109]. Although CX3CR1^hi^-resident macrophages retain their anti-inflammatory signatures [27,97,110], which suggests that they might play a role in immunomodulation, the differentiation of CX3CR1 monocytes is delayed. This relative depletion leads to subsequent inflammation. Reduced CX3CR1 is also associated with increased gut permeability [111,112,113,114].

The CCR2-CCL2 axis also governs monocyte migration in humans. Classical monocytes express CCR2 [115], and an upregulation of its ligand, CCL2, has been found in the livers and intestines of AUD patients [104]. Furthermore, radio-labeled CD14^hi^ monocytes have been shown to migrate to inflamed tissue of IBD mucosa [116]. In a healthy gut, resident macrophages and other lymphocytes contribute to the recruitment of monocytes by secreting ligands such as CCL2, CCL7, CCL8, etc. [117]. Nonetheless, it is important to note that CCR2 is not the only receptor, nor does it govern monocyte migration in all contexts, as the recruitment of Ly6C^hi^ monocytes and the accumulation of its progeny are unaffected in *H. hepaticus*-induced colitis [118]. CCR1 and CCR5 have specialized roles in monocyte recruitment, as demonstrated by transmigration studies and in some cases of inflammation [119,120]. A recent animal study demonstrated that levels of hepatic-monocyte-derived macrophages increased in mice chronically given alcohol due to the increase in both Ly6C^hi^ and Ly6C^lo^ subsets, the migration of which is mediated by the CX3CR1-CX3CL1 axis [113,121].

The CX3CR1^int^ Mϕ are localized beneath the epithelial barrier. This positioning, coupled with their high phagocytic capacity, indicates that they are designed to capture and clear invading microbes or pathogens as well as apoptotic and senescent cells. When homeostasis is perturbed by inflammation or infection caused by alcohol abuse, CD14+ monocytes and their CX3CR1^int^ derivatives accumulate in large numbers and display enhanced pro-inflammatory characteristics. They may also be able to directly sample the luminal contents through their extending processes between the cells of the intestinal epithelium. CX3CR1^int^ Mϕ also constitutively produce interleukin-10 (IL-10), which facilitates the secondary expansion of regulatory T cells in the mucosa and may also condition newly arrived monocytes. However, in the setting of alcohol-associated liver disease, the activation of monocytes/macrophages is changed due to a different microenvironment.

### 6.2. Macrophage Activation in Alcohol-Induced Liver Disease

One of the important elements of disease development and progression is the macrophage. The number of macrophages in the liver is higher during the early stage of fatty liver disease as well as in end-stage hepatitis and cirrhosis [122]. In one study, it was observed that patients with alcohol-associated steatohepatitis had hepatic macrophages expressing both anti-inflammatory and pro-inflammatory cytokines, indicating that the macrophage can adopt both phenotypes simultaneously [116]. By contrast, the number of intestinal macrophages in AUD patients is reduced [9]. Yet, the reason for the depletion of intestinal macrophages and their role in liver disease progression remain unknown.

The literature has shown that the expression levels of inflammatory genes in hepatic macrophages are upregulated [123]. Indicators of macrophage activation, such as CD58, neopterin, and leukocyte-function-associated antigen 3, as well as cytokines such as IL-6, IL-8, IL-18, and chemokines, are all elevated in ALD patients [124,125,126]. Research on a murine model has suggested that acute and chronic exposure to alcohol elevate TNF-α, IL-6, MCP1, and ROS, consistent with the activation of hepatic macrophages [99,127]. Gadolinium-chloride-induced macrophage depletion has been shown to lead to attenuated liver inflammation [128,129]. Murine KCs subjected to chronic alcohol exposure are sensitive to LPS and show an increased immune response to it [102,103]. One study also found that chronic alcohol exposure can lead to an accumulation of infiltrated monocytes in the liver [108]. Intriguingly, circulating monocytes from ALD patients produce more TNF-α and are more sensitive to LPS [127,130] compared to healthy subjects. Liver-infiltrating monocytes fall under two groups: one is composed of classical monocytes, and the other is composed of non-classical monocytes. Classical-monocyte-derived macrophages tend to exhibit a pro-inflammatory phenotype, and non-classical-monocyte-derived macrophages tend to have a pro-fibrogenic phenotype [108]. Apart from activation by macromolecules, miRNA also plays a role as a modulator of macrophage activation and has been studied extensively in recent years.

### 6.3. Micro RNA as a Modulator of Alcohol Response in Macrophages

miRNAs are small non-coding RNAs, mostly transcribed from DNA into primary miRNA and later processed into precursor miRNA and mature miRNA. miRNA plays an important role in the regulation of inflammatory pathways and macrophage functions in ALD [49]. Some studies have shown that a set of miRNAs can modulate the alcohol-induced hyperinflammatory response though the inhibition of TLR pathways in the liver and intestines [131,132]. Exposure to alcohol can upregulate the Kupffer-cell-derived miRNAs miR-155, miR-217, and miR132 [133,134,135]. miR-155 has an inhibitory effect on the TLR4 pathway, which in turn results in hyper-sensitivity to LPS and the upregulation of TNFα in the liver [133]. Moreover, miR-155 upregulates the release of pro-inflammatory Kupffer-cell-derived cytokines such as TNF-α and IL-1β and downregulates IL-10 in the meantime [136,137]. miR-132, on the other hand, upregulates the expression of TGF-β, IL-1β, and MCP-1 in hepatic macrophages [138]. This upregulation of pro-inflammatory cytokines shows a close tie to chronic liver disease and inflammation, as the levels of these cytokines are increased in murine models of alcohol intake. In a human study, higher levels of miR-132 were associated with fibrosis and cirrhosis [138]. Other miRNAs can also contribute to the downregulation of the monocyte/macrophage cytokine response. miR-217, miR-291b, and miR-181b-3b were shown to be responsible for the regulation of TFG-β in Kupffer cells in murine models and negatively regulate TLR signaling pathways [134,139,140]. Apart from altering cytokine and chemokine secretion, alcohol-induced miRNAs can also affect the function of monocytes/macrophages by inducing oxidative stress and phagocytosis. miR-214 expression in the liver is exacerbated by alcohol-associated oxidative stress [141]. Overexpression of miR-212 has been found in intestinal biopsies of ALD patients with intestinal barrier dysfunction and in whom ZO-1 was downregulated [142]. Upregulated miR-217 also leads to increased ROS production in KCs [134]. LPS-stimulated KCs from rats given alcohol show an increased amount of miR-125a-5p, which inhibits TLR4 signaling and leads to the polarization of macrophages to a pro-fibrogenic phenotype [143]. An early study on non-human primates showed downregulation of miR-27a in peripheral blood mononuclear cells (PBMCs), a process that drives the polarization of macrophages towards a pro-inflammatory state by secreting IL-10. This in turn indicates that alcohol can induce a phenotypic change in macrophages [144].

Under normal conditions, circulating monocytes will respond to chemokines during inflammation and subsequently penetrate tissue and differentiate into macrophages. In the steady state, their responses are tightly regulated, and these cells will further polarize into different subsets of macrophages by responding to different cytokines and miRNAs to properly attenuate the inflammation in the gut and liver. However, under pathological conditions, this delicate homeostasis is disrupted. Both monocyte-derived macrophages in the intestine as well as Kuppfer cells in the liver undergo phenotypic changes. Although all aspects of this process are not fully understood, chronic alcohol exposure seems to drive the polarization of macrophages towards a pro-inflammatory and pro-fibrotic phenotype, which ultimately contributes to tissue damage and disease progression.

## 7. Limitation of Animal Models in ALD Studies

Murine models have been widely used in many fields. However, murine models are suboptimal tools for ALD research due to fundamental differences between mice and humans. For instance, the basal metabolism rate per gram of body weight in mice is seven times higher than that in humans [145]. Specifically, the alcohol metabolism rate is five times higher than that in humans [146]. There is also a huge difference regarding immune cells. The blood of mice has a greater preponderance of lymphocytes, whereas human blood predominantly contains neutrophiles [147]. The human immune system is considered resistance-induction-dominant, which means it tends to directly deactivate and eliminate invading pathogens, whereas the mouse immune system tends to exhibit tolerance induction and not be responsive to a specific pathogen or set of pathogens [148]. Moreover, during alcohol-induced inflammation, the human immune system is challenged by TLR ligand/agonists such as LPS and PGN, whereas the murine immune system hardly responds to TLR ligands [149,150,151]. Additionally, mice respond to IFNγ by releasing 18 different kinds of p47 immunity-related GTPase (IRG) proteins to induce a series of cellular activities to clear out the pathogens [152]. By contrast, humans only express two kinds of IRG proteins, neither of whose activity is induced by IFNγ. One of them does not even display any antimicrobial properties apart from participating in elimination of mycobacteria by macophages [149,153]. As mentioned before, the main challenge in this regard is the lack of adequate markers with which to properly identify macrophage populations and their functionality. Many markers in mice, such as arginase 1 and Fizz1, are not found in humans, whereas human markers such as fibrinoligase and Ykl39 are not expressed in mice [154]. Additionally, cytokines such as iNOS display a striking difference. In murine macrophages, iNOS plays an important role in NO and L-citrulline synthesis [155], whereas human macrophages do not produce any iNOS, and NO is not synthesized by them [156,157]. Murine gut microbiota compositions are also different from those of humans, with humans having a higher Firmicutes-to-Bacteroidetes ratio (F/B ratio) [158]. Additionally, both humans and mice carry distinct species, which makes the composition largely different from one specimen to another [159]. Ad libitum alcohol feeding in mouse models are the most common models for the study of ALD. In one study, after eight weeks to up to 70 weeks of alcohol feeding, no significant change in mortality was observed [160]. In most cases, ab libitum feeding can induce a liver injury with elevated levels of aspartate aminotransferase and alanine aminotransferase, but no liver fibrosis would be observed [160,161], and therefore this model is unsuitable for reproducing the complexity of human disease. Although studies on murine models have given us a rough understanding of the innate immune system and macrophages in the pathogenesis of ALD, this research is hard to transpose to humans due to the different physiology and immune responses to alcohol between mice and humans. More in-depth clinical studies are needed to acquire a better understanding of ALD progression and the correlation with macrophages. Alternatively, PBMCs have emerged as a novel target for the study of pathogenesis and as a therapeutic target for alcohol-associated liver disease, specifically targeting mitochondrial oxidative function and modulating the immune response to alcohol-induced inflammation in ALD [18,19]. Liver explant studies have shown a correlation between monocyte chemotactic protein levels and MELD scores primarily in non-alcohol etiologies [162]. Nonetheless, macrophage accumulation is also observed in liver explants [163]. However, these observations in liver explants indicate end-stage liver disease, which cannot be extrapolated to the condition of chronic alcohol consumption in less advanced disease stages.

## 8. Therapeutic Interventions and Future Research

In ALD, the current standard treatment consists of alcohol abstinence and the use of medications such as disulfiram, naltrexone [164,165], and acamprosate [166] to eventually prevent relapses. It is not uncommon to observe a disrupted microbiota and increased intestinal permeability in ALD patients [9]. There are treatments available that can restore eubiosis in the gut, such as dietary regimens, probiotics, and prebiotics. Some clinical studies have shown that in ALD patients, probiotic treatment reduces the levels of aspartate aminotransferase, alanine transferase, lipopolysaccharides, and TNFα [167]. However, these treatments have not been proven to have long-term benefits. Apart from medical treatment, fecal microbiome transplantation has effectively been used to decrease gut dysbiosis and reduce gut permeability in animal models [166]. Also, in recent years, clinical trials have suggested that a fecal microbiome transplant can improve gut immunity and alleviate gut dysbiosis in humans (see https://clinicaltrials.gov/ct2/show/NCT02862249 accessed on 30 September 2019, https://clinicaltrials.gov/ct2/show/NCT02400216, and https://clinicaltrials.gov/ct2/show/NCT02496390 accessed on 11 December 2018). Although the gut microbiota might be a promising therapeutic and diagnostic target in AUD and ALD, convincing evidence of a benefit for humans is still lacking. According to a recent human study, an upregulation of TLR-2 signaling in circulating monocytes and liver-infiltrating monocytes is associated with inflammation in AUD and ALD. Hence, inhibiting TLR-2 could be a therapeutic target for reducing systemic inflammation and mitigating liver damage in ALD. In addition, the IL-1β- and IL-8-driven inflammatory response in monocytes is related to the activation of the NLRP-3 inflammasome, indicating that the inflammasome is another potential therapeutic target [104]. It is also conceivable that inflammation and monocyte recruitment could be mitigated by using chemokines inhibitors, as suggested in a recent human study targeting myometrial myocyte–macrophage communication [160]. In addition, targeting CSF1 has already been applied in cancer therapy [168]. It might be a potential alternative approach to, for example, alleviating liver inflammation in ALD. Macrophage-targeted therapy is often used in many cancer therapies. In recent studies, it has been pointed out that switching hepatic macrophages from a pro-inflammatory to pro-fibrogenic phenotype can alleviate inflammation and promote tissue repair in individuals with non-alcoholic fatty liver disease (NAFLD). This phenomenon is mainly driven by the upregulation of anti-inflammatory cytokines related to tissue repair and regeneration [169,170,171]. Currently, no such investigation has been conducted in relation to ALD. Even though we have gained some insight into the relationship between alcohol and the gut microbiota, many questions remain open. A better understanding of how alcohol affects bacterial communities, bacterial translocation into blood or the lymphatic system, the stimulation of the vagus nerve, the endocrine system, and alterations of immune systems will allow us to fill in the missing pieces in the pathophysiology of AUD and its remote organ damage such as in the liver. Ongoing debate about the potential role of tissue-specific macrophages is raising controversies in the field of immunology. Recently, demographic data regarding adipose tissue macrophages were investigated in an NAFLD study, demonstrating transcriptomic differences between adipose-resident vascular macrophages and metabolically active macrophages. In animal research, a series of studies have investigated the subsets of macrophages and their roles regarding spatial distribution. Additionally, resident macrophages are not exclusively embryonically derived: they can also be replenished by circulating monocytes [166]. However, little is known about the subsets of macrophages in the intestines and liver and their metabolic and functional differences in terms of spatial distribution in humans. This gap needs to be filled by more in-depth studies on humans, given the fundamental difference between humans and animal models in alcohol research.

Despite our growing understanding of how macrophages are an important member of the mononuclear phagocyte system and the innate immune system, little is known about the phenotypes and the different functions of residential macrophages and monocyte-derived macrophages. Furthermore, better markers need to be identified in order to gain a better understanding of the functions and phenotypes of different subsets of intestinal and hepatic macrophages. More studies about human tissue macrophages should be carried out using cutting-edge technology such as scRNA sequencing and omic technology. Additionally, since findings obtained using animal models are difficult to translate into clinical implications, better in vitro or ex vivo systems need to be developed, such as organoids, to study the pathogenesis of alcohol-associated liver disease and the effect of alcohol on the intestines.

## Figures and Tables

**Figure 1 cells-14-00207-f001:**
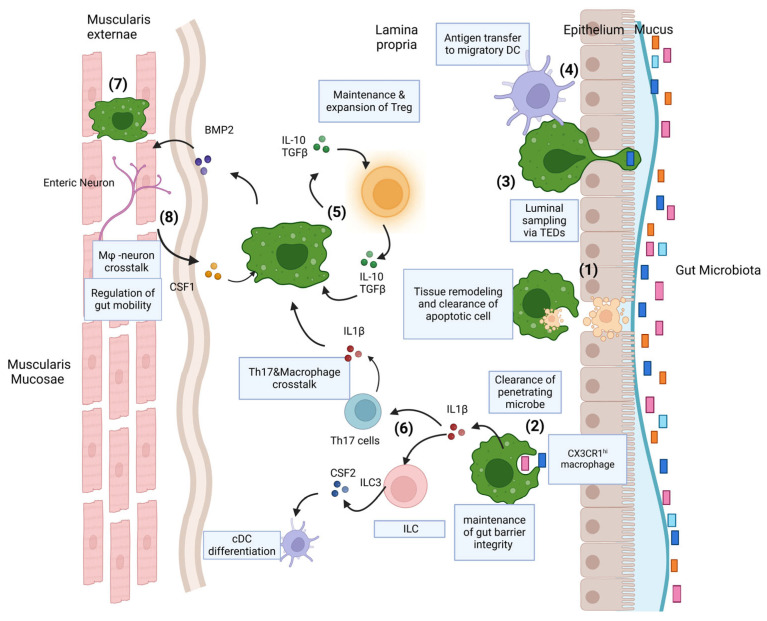
Homeostatic functions of intestinal macrophages. Intestinal lamina propria (LP) macrophages are highly phagocytic, and they are responsible for (1) clearing apoptotic and senescent epithelial cells. LP macrophages’ position under the epithelial barrier and highly phagocytic nature indicate they are ideally placed (2) to capture and clear any pathogens that penetrate the barrier. They may also (3) initiate cellular processes across the epithelial barrier to sample luminal contents. Macrophages can (4) transfer acquired antigens to migratory dendritic cells (DCs) for presentation to T cells in draining mesenteric lymph nodes. (5) Through their production of immunoregulatory cytokines, such as IL-10 and TGFβ, they maintain and facilitate secondary expansion of regulatory T cells (Tregs) locally in the LP, which in turn regulate the inflammatory statuses of macrophages. In a similar manner, (6) they support Th17 cells and ILC3s through their production of IL1β, which is induced by exposure to the microbiota or its derivatives. (7) Macrophages are also present in deeper layers of the gut wall, including the submucosa and muscularis externae. Muscularis macrophages participate in (8) bidirectional crosstalk with sympathetic neurons of the enteric nervous system and influence gut motility.

**Figure 2 cells-14-00207-f002:**
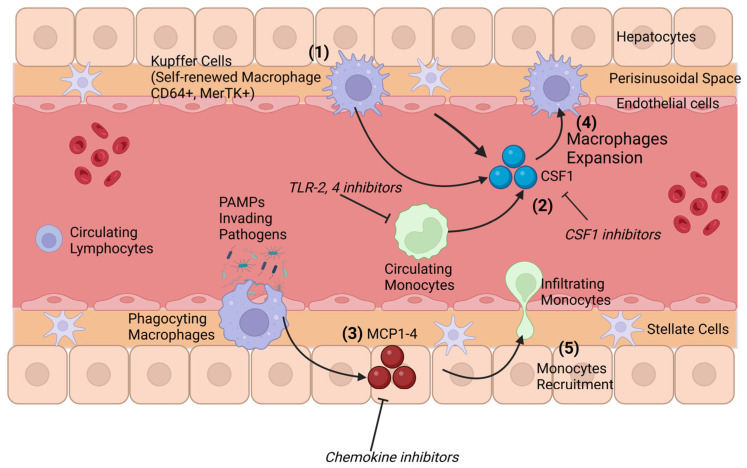
Liver macrophage heterogeneity. Hepatic macrophages are composed of different populations. (1) The most abundant one is composed of embryonically derived Kupffer cells (KC), which reside in liver sinusoids. In chronic alcohol abuse, (2) CSF1 and (3) MCP1-4 levels are elevated because of monocytes and macrophage activation. This in turn promotes the additional (4) macrophages’ expansion and (5) the recruitment of monocytes into the tissue, which might contribute to excessive systemic or local inflammation. Monocyte-derived macrophages can also adopt KCs’ phenotype and become long-lived tissue-resident macrophages. The potential therapeutic targets indicated in italic in the figure above have not been tested in the alcohol setting.

**Figure 3 cells-14-00207-f003:**
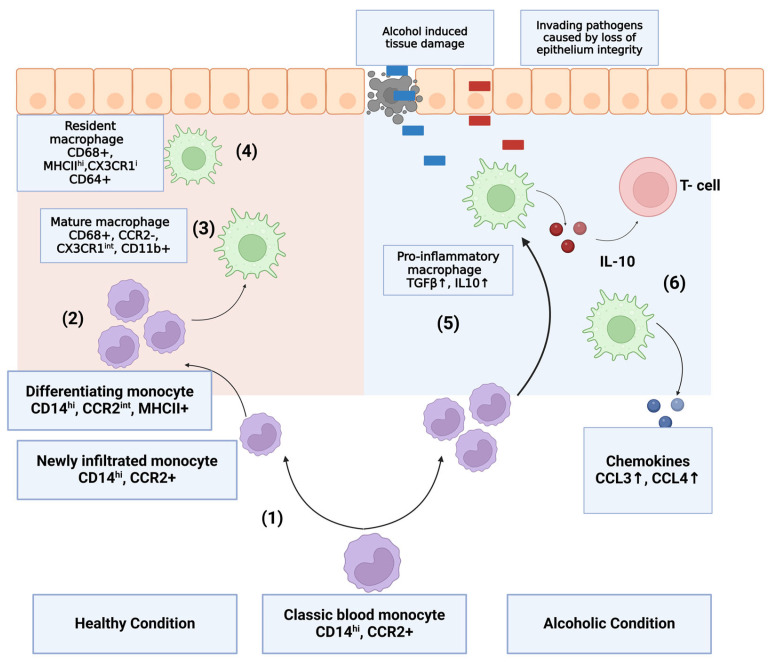
Under healthy conditions, (1) CD14+ monocytes constitutively enter the intestinal mucosa and differentiate into (3) mature CX3CR1^int^ and CD68^+^ macrophages (Mϕ) through a series of (2) short-lived CCR2^int^ MHC-II^+^ intermediates. (4) CX3CR1^int^ Mϕ macrophages are localized beneath the epithelial barrier. This positioning, coupled with their high phagocytic capacity, indicates that they are designed to capture and clear invading microbiome or pathogens as well as apoptotic and senescent cells. When homeostasis is perturbed by inflammation or infection caused by alcohol abuse, (5) CD14+ monocytes and their CX3CR1^int^ derivatives accumulate in large numbers and display enhanced pro-inflammatory characteristics. (6) CX3CR1^int^ Mϕ macrophages also constitutively produce interleukin-10 (IL-10), which facilitates the secondary expansion of regulatory T cells in the mucosa and may also condition newly arrived monocytes along with upregulated chemokines CCL3 and CCL4.

## Data Availability

Not applicable.

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
