# Peer review of "Origin, Function, and Implications of Intestinal and Hepatic Macrophages in the Pathogenesis of Alcohol-Associated Liver Disease"

_cells, 2025, doi:10.3390/cells14030207_

Round 1
Reviewer 1 Report
Comments and Suggestions for Authors
Hu et al. have written this review article entitled "Origin, Function, and Implication of Intestinal and Hepatic 2 Macrophage in Pathogenesis of Alcohol Associated Liver Disease". The authors have done a commendable job in capturing the literature on intestinal and hepatic macrophages in the pathogenesis of alcohol-associated liver disease (ALD). Especially, the figures are excellent which capture the content nicely. However, there is a minor limitation of connection with human / translational work, although the authors have outlined issues and limitations of this angle. There are obvious issues with studying intestinal and hepatic macrophages in human samples from ALD patients, the authors may like to embark upon available literature and studies on peripheral blood mononuclear cells (PBMC) and hepatic macrophages from explants to study the pathogenesis of ALD and derive biomarkers for diagnosis and prognosis in patients with ALD. This is relevant given the recognition that human phenotype of ALD does not match with animal models of ALD. This is also relevant for drug development in ALD.
Author Response
Dear Reviewer,
Thank you for your kind comment and evaluation for our manuscript and the constructive comments and suggestions. We have considered each point raised and have addressed them in the details below. Changes has been maked in section 8. therapeutic interventions and future research t as outlined below
-PBMC have emerged as a novel target for the study of pathogenesis and therapeutic target for alcohol-associated liver disease, specifically target at mitochondrial oxidative function and modulating the immune response to alcohol liver disease induced inflammation.
-Liver explant study has shown correlation of monocyte chemotactic protein level and MELD score. Nonetheless, macrophage accumulation is also observed in liver explant. However, these observations in liver explant often indicate terminal stage liver disease which hardly resemble the condition of chronic alcohol consumption.
Reviewer 2 Report
Comments and Suggestions for Authors
The authors present a comprehensive review of the role of intestinal and hepatic macrophages in alcohol-associated liver disease (ALD), with a focus on their origins, functional dynamics, and implications in disease progression. While the manuscript addresses a significant and timely topic, there are areas for improvement that would strengthen the review's impact and clarity. Below are my comments:
Major Comments:
1. While the manuscript is generally well-organized, certain sections lack a clear flow. For example, the section on macrophage dynamics in the intestine (Section 3) is overly detailed, with some redundancy regarding the influence of microbiota. Condensing this section would improve readability. Additionally, the transition from macrophage origins to their functional roles in ALD could be more seamless. Adding a summary paragraph at the end of each major section would help tie the narrative together and improve cohesion.
2. The manuscript reiterates well-known concepts, such as TLR4 signaling and macrophage activation in ALD, without providing a novel synthesis. To enhance the impact, I recommend emphasizing recent advances or controversies in the field (e.g., the roles of tissue-specific macrophage subsets). Furthermore, integrating discussions on how macrophages interact with other immune cells and how gut microbiota-derived metabolites influence macrophage polarization would add depth and novelty to the review.
3. The figures are helpful but could benefit from more detailed annotations, particularly Figure 2. Including additional diagrams illustrating therapeutic strategies targeting macrophages would make the review more informative and visually engaging.
4. Section 7, "Limitations of Animal Models in ALD Studies," does not align closely with the primary focus of the review. I suggest removing this section or significantly revising it to directly relate the limitations of animal models to the study of macrophage roles in ALD.
Minor Comments:
1. Use consistent terminology throughout (e.g., "alcohol-associated liver disease" vs. "alcoholic liver disease"). The former is the preferred term in recent literature.
2. Professional English editing is recommended for minor grammatical errors.
Author Response
Dear Reviewer,
Thank you for your kind comment and evaluation for our manuscript and the constructive comments and suggestions. We have carefully reconsidered each point raised and made additional changes as outlined below. Changes outlined in our response have been marked in the manuscript.
Comment 1: However, it is somewhat difficult to read as the logical structure is not always evident and links between different sections are not well phrased. This concerns all sections including section 3.
Response: The introduction of section 3 has been considerably shortened. Section 3.2 “The impact of an altered gut microbiota in AUD now has been deleted”. The most relevant information of this section has been integrated into the remaining sections, redundant information concerning microbiota has been removed. We also tried to ameliorate the coherence between the different paragraphs.
Comment 2: The manuscript reiterates well-known concepts, such as TLR4 signaling and macrophage activation in ALD
Response: we added the following sentences to section 6.1:
Apart from the existing evidence of the role of TLR-4 in ALD, it has been demonstrated that TLR-2 plays an important role in monocyte activation during the inflammatory response induced by alcohol. The activation of TLR-2, which was dependent on peptidoglycan but not LPS, is associated with the exacerbation of liver damage.
We also added information about TLR-2 as a potential therapeutic target in the section 8 (see below)
Comment 3: To enhance the impact, I recommend emphasizing recent advances or controversies in the field (e.g., the roles of tissue-specific macrophage subsets)
Response: We added the following paragraphs to section 3.1 and the discussion respectively
Research based on chronically alcohol fed mice also indicates that CD11c+ macrophages are concentrated close to the tips of the villi and express a pro-inflammatory profile, whereas CD206+ macrophages are located at the base of the crypt fulfill the function of tissue macrophage.
On-going debate about the potential role of tissue-specific macrophages raises controversies in the immunology field. Recently, the demographic data of adipose tissue macrophages were investigated in a NAFLD study, demonstrating the transcriptomic difference between adipose resident vascular macrophage and metabolic active macrophages. In animal research, a series of research has been done in investigating the subsets of macrophages and their roles regarding the spatial distribution. Additionally, resident macrophages are not exclusively embryonically derived but can also be replenished by circulating monocytes. However, little is known about the subsets of macrophages in intestine and liver and their metabolic and functional differences in terms of spatial distribution in humans.
Comment 4: Furthermore, integrating discussions on how macrophages interact with other immune cells...
Response: Section 3.2 is already dedicated to the interactions with macrophage and other type of cells especially with immune cells.
We further added the following paragraphs discussion the role of IL-10 in the crosstalk between T cells and macrophages
- IL-10 does not only limit the inflammatory state of macrophages but also regulates the killing of intracellular bacteria by those cells. Human-induced pluripotent stem cell–derived macrophages without a functional IL-10R display impaired bacterial killing due to dysregulated PGE2 production
- On the other hand, T cell derived IL-10 is required for the regulation of intestinal homeostasis by a T cell-macrophage IL-10 axis. Indeed, appropriate IL-10 signalling levels are important for regulating different macrophage functions
Additionally, we also added the following sentences to section 3.3 to provide additional information of the crosstalk between macrophages and enteric neurons.
- Besides, intestinal macrophages might also serve as signal transmitting cells in neuro-immune-stem cell regulatory circuits, modulating intestinal stem cell self-renewal by upstream enteric serotonergic neurons. This process implicates the neurotransmitter serotonin production in enteric neurons which can be promoted by gut microbiota metabolite valeric acid.
Comment 5: Figure 2. Including additional diagrams illustrating therapeutic strategies targeting macrophages
Response: We revised the figures and the figure legends. We included more detailed annotations and removed the elements that are not visible in the figures. Potential therapeutic targets have been added to Figure 2.
Comment 6: Therapeutic strategies targeting macrophages would make the review more informative and visually engaging.
Response: We added the following sentences concerning the potential therapeutic strategies in the section 8 “Therapeutic interventions and future research”:
- According to a recent human study, the upregulation of TLR-2 signaling in circulating monocytes and liver infiltrating monocytes was associated with inflammation in AUD and ALD. Hence, inhibiting TLR-2 could potentially be a therapeutic target to reduce systemic inflammation and mitigate liver damage in ALD. In addition, the IL-1β and IL-8 driven inflammatory response in monocytes was related to the activation of the NLRP-3 inflammasome, which poses the inflammasome as a potential therapeutic target, as well. It is also conceivable that inflammation and monocyte recruitment could be mitigated by using chemokines inhibitors as suggested in a recent human study targeting myometrial myocyte-macrophage. In addition, targeting CSF1 has already been applied in cancer therapy. it might be a potential alternative approach, for example, of alleviating liver inflammation in ALD.
We also highlighted these strategies in the new figure 2.
Other specific comments have been addressed in the previous revision.
Reviewer 3 Report
Comments and Suggestions for Authors
The challenge here is translational research to humans. Subsequently the challenge is tailoring treatment based on the pathogenesis in a specific patient. There is a wide spectrum of contributing phenomena in a given patient. Furthermore we know that 50% of current concepts will change in 10 years. Yet alcoholic liver disease is a major neglected area for successful treatment. At this point in time is your group conducting a speciific factor in alcoholic liver disese which will prove to be efficacious in treatment?
Author Response
Dear Reviewer,
Thank you for your kind comment and evaluation for our manuscript and the constructive comments and suggestions.
With regard to your question, in the past we tried to modulate the composition of gut microbiome with inulin (PMID: 35490461). However, the result is not in favor of improvment of liver condition in the patients. At moment, we are trying to modulate systemic inflammation in a still ongoing study.
Round 2
Reviewer 2 Report
Comments and Suggestions for Authors
Thank you for your response to my initial review.
While I appreciate the efforts made to address the concerns raised, I am disappointed by the lack of care and rigor reflected in the revisions and responses. The overall quality of the revisions and the authors’ response suggests a lack of thoroughness and attention to detail. The revisions feel superficial, with inadequate effort made to engage with the review comments meaningfully.
Author Response
Dear Reviewer,
Thank you for your kind comment and evaluation for our manuscript and the constructive comments and suggestions. We have carefully reconsidered each point raised and made additional changes as outlined below. Changes has been marked to the manuscript as outlined in our response
Comment 1: While the manuscript is generally well-organized, certain sections lack a clear flow. For example, the section on macrophage dynamics in the intestine (Section 3) is overly detailed, with some redundancy regarding the influence of microbiota. Condensing this section would improve readability.
Response: the introduction of this section has been considerably shortened, section 3.2 The impact of an altered gut microbiota in AUD now has been deleted. The most relevent information of this section has been integrated into the remaining sections, redundent information concerning microbiota has been removed.
Comment 2:The manuscript reiterates well-known concepts, such as TLR4 signaling and macrophage activation in ALD
Response: we added the following sentences to section 6.1:
Apart from the existing evidence of the role of TLR-4 in ALD, it has been demonstrated that TLR-2 plays an important role in monocyte activation during the inflammatory response induced by alcohol. The activation of TLR-2, which was dependent on peptidoglycan but not LPS, is associated with the exacerbation of liver damage.
we also added information about TLR-2 as a potential therapeutic target in the section 8 (see below)
Comment 3: To enhance the impact, I recommend emphasizing recent advances or controversies in the field (e.g., the roles of tissue-specific macrophage subsets)
Response: We added the following paragraphs to section 3.1 and the discussion respectively
Research based on chronically alcohol fed mice also indicates that CD11c+ macrophages are concentrated close to the tips of the villi and express a pro-inflammatory profile, whereas CD206+ macrophages are located at the base of the crypt fulfill the function of tissue macrophage.
On-going debate about the potential role of tissue-specific macrophages rises controversies in the immunology field. Recently, the demographic data of adipose tissue macrophages were investigated in a NAFLD study, demonstrating the transcriptomic difference between adipose resident vascular macrophage and metabolic active macrophages. In animal research, series of research has been done in investigating the subsets of macrophages and their roles regarding the spatial distribution. Additionally, resident macrophages are not exclusively embryonically derived but can also be replenished by circulating monocytes. However, little is known about the subsets of macrophages in intestine and liver and their metabolic and functional differences in terms of spatial distribution in humans.
Comment 4: Furthermore, integrating discussions on how macrophages interact with other immune cells...
Response: Section 3.2 is now dedicated to the interactions with macrophage and other type of cells especially with immune cells.
Additionally, we also added the following sentences to section3.3 to provide additional information of the crosstalk between macrophages and eneric neuron.
Besides, intestinal macrophages might also serve as signal transmitting cells in neuro-immune-stem cell regulatory circuit, modulating intestinal stem cell self-renewal by upstream enteric serotonergic neurons. This process implicates the neurotransmitter serotonin production in enteric neurons which can be promoted by gut microbiota metabolite valeric acid
Comment 5: Figure 2. Including additional diagrams illustrating therapeutic strategies targeting macrophages
Response: Figure 2 annotation and legend have been revised. potential therapeutic targets have been added.
Comment 6:therapeutic strategies targeting macrophages would make the review more informative and visually engaging.
Response: We added the following sentences concerning the potential therapeutic strategies in the section 8 Therapeutic interventions and future research:
According to a recent human study, the upregulation of TLR-2 signaling in circulating monocytes and liver infiltrating monocytes was associated with inflammation in AUD and ALD. Hence, inhibiting TLR-2 could potentially be a therapeutic target to reduce systemic inflammation and mitigate liver damage in ALD. In addition, the IL-1β and IL-8 driven inflammatory response in monocytes was related to the activation of the NLRP-3 inflammasome, which pose the inflammasome as a potential therapeutic target, as well. It is also conceivable that inflammation and monocytes recruitment could be mitigated by using chemokines inhibitors as suggested in a recent human study targeting myometrial myocyte-macrophage. In addition, targeting CSF1 has already been applied in cancer therapy. it might be a potential alternative approach, for example, of alleviating liver inflammation in ALD.
Other specific comments have been addressed in the previous revision.
Round 3
Reviewer 2 Report
Comments and Suggestions for Authors
No further comments.